# Usability of a Psychotherapeutic Interactive Gaming Tool Used in Facial Emotion Recognition for People with Schizophrenia

**DOI:** 10.3390/jpm11030214

**Published:** 2021-03-17

**Authors:** Roberto Pablo González, Ingrid Tortadès, Francesc Alpiste, Joaquín Fernandez, Jordi Torner, Mar Garcia-Franco, José Ramón Martin-Martínez, Sònia Vilamala, Maria Jose Escandell, Emma Casas-Anguera, Gemma Prat, Susana Ochoa

**Affiliations:** 1Universidad Politécnica de Cataluña LAM-UPC, BARCELONA TECH Campus Diagonal Nord, Edifici Ω Omega. C. Jordi Girona, 1-3, 08034 Barcelona, Spain; rpablog@hotmail.com (R.P.G.); francesc.alpiste@upc.edu (F.A.); jfernandez@ege.upc.edu (J.F.); jordi.torner@upc.edu (J.T.); 2Parc Sanitari Sant Joan de Déu CIBERSAM, Sant Boi de Llobregat, 08830 Barcelona, Spain; itortades@sjdhospitalbarcelona.org (I.T.); mmgarcia@pssjd.org (M.G.-F.); jrmartin@pssjd.org (J.R.M.-M.); svilamala@pssjd.org (S.V.); mjescandell@pssjd.org (M.J.E.); ecasas@pssjd.org (E.C.-A.); 3Divisió de Salut Mental, Fundació Althaia, Carrer del Dr. Joan Soler Núm. 1-3, Manresa, 08243 Barcelona, Spain; gprat@althaia.cat; 4Unitat de Recerca, Parc Sanitari Sant Joan de Déu, Dr. Antoni Pujadas, 42, Sant Boi de Llobregat, 08830 Barcelona, Spain

**Keywords:** facial, emotion, recognition, tool, schizophrenia, social cognition, gaming

## Abstract

The objective of the study was to test the usability of ‘Feeling Master’ as a psychotherapeutic interactive gaming tool with LEGO cartoon faces showing the five basic emotions, for the assessment of emotional recognition in people with schizophrenia in comparison with healthy controls, and the relationship between face affect recognition (FER), attributional style, and theory of mind (ToM), which is the ability to understand the potential mental states and intentions of others. Nineteen individuals with schizophrenia (SZ) and 17 healthy control (HC) subjects completed the ‘Feeling Master’ that includes five basic emotions. To assess social cognition, the group with schizophrenia was evaluated with the Personal and Situational Attribution Questionnaire (IPSAQ) for the assessment of attributional style and the Hinting Task (ToM). Patients with SZ showed significant impairments in emotion recognition and their response time appeared to be slower than the HC in the recognition of each emotion. Taking into account the impairment in the recognition of each emotion, we only found a trend toward significance in error rates on fear recognition. The correlations between correct response on the ‘Feeling Master’ and the hinting task appeared to be significant in the correlation of surprise and theory of mind. In conclusion, this study demonstrated that the ‘Feeling Master’ could be useful for the evaluation of FER in people with schizophrenia. These results sustain the notion that impairments in emotion recognition are more prevalent in people with schizophrenia and that these are related with impairment in ToM.

## 1. Introduction

### 1.1. Impairment in Emotion Recognition in Schizophrenia

Emotional recognition deficit is one component of social cognition. Social cognition refers to a cognitive capability applied to social situations, consisting of a group of cognitive processes, which allows the effective use of social conventions in real-world situations [1,2]. Apart from emotional recognition, social cognition comprises theory of mind, attributional style, and social perception [3,4,5]. Theory of Mind (ToM) consists of the ability to understand the potential mental states and intentions of others, while attributional style includes a general tendency to explain personally significant events [6]. Social perception consists of the understanding of social clues; however, there is no instrument in Spanish for the assessment of this dimension.

Emotional recognition deficits are well documented as being prominent among patients with schizophrenia [7,8,9,10,11,12,13,14,15]. Specifically, people with schizophrenia present higher deficits in the identification of negative emotions such as angry faces, but not in the recognition of happy faces [16,17].

### 1.2. Relation between Emotional Recognition and Other Variables (Clinical and Social)

Relationships between emotional recognition deficit and the presence of psychotic symptoms have been reported [18,19]. In addition, cognitive functioning and social functioning have been related to emotional recognition [12,20,21]. According to Couture et al. [21], impairments in different domains of social cognition, including emotional recognition and ToM, are correlated with decreaced social functioning. Social cognitive deficits have a strong correlation with real-world social outcome [4,20]. In this way, an improvement in emotional recognition in patients with schizophrenia could be related with an improvement in their clinical symptoms, social functioning, and social cognition.

### 1.3. Assessment of Emotional Recognition

Traditionally, the clinical assessment of emotional recognition has been based on paper and pencil tools. Although the study of emotional recognition has been considerably well explored, there are few validated instruments for the clinical assessment of this construct. The social cognition assessed by the MSCEIT, included in the MATRICS battery (measurement and treatment research to improve cognition in schizophrenia), includes a part of emotional recognition based on ambiguous expressions in faces [22]. Cassé-Perrot C. et al. [23] developed the Emotional State Questionnaire (ESQ), which includes recognition, expression, internal emotional experience, and social context. Another instrument developed for the assessment of emotional recognition is the Emotional Recognition Index (ERI) by Scherer & Scherer [24]. Most of the instruments are paper and pencil tools or the computerized version of the instrument including only the total score of the test, but no other interactive information such as the reaction time to response, the possibility of modifying the order of stimuli, or the exposure time.

### 1.4. Informatics Application Tools for the Assessment

In the past decades, we have witnessed a great development and expansion of the computer-assisted therapy programs and its possible potential in clinical settings [25,26]. The past research suggests that computer assessment and therapy tools offer the possibility to increase the cost-effectiveness of modern psychotherapeutic interventions by reducing therapist contact time and increasing patient engagement in therapeutic activities in non-clinical environments [25,27,28].

The impact of these new technology tools will be more prominent as they become more available to evaluate and to elaborate new treatments for individuals with a mental illness. Applications made for smartphones and tablets could be useful in the treatment of patients with a mental illness [29].

Furthermore, the acceptance of computer games has been increasing very rapidly in the last 10 years and they have been largely used by children, adolescents, and adults. Mental health therapists have been searching how the use of these games could be a complement to a traditional treatment method where the key to success is that they must be well-designed [30]. Until now, there is little evidence of the effectiveness of the use of new technologies in the treatment of mental health illnesses in children and adults. Research on this topic has been limited to autism, with more attention and more interest in cartoon characters compared with human faces [31,32,33] and greater improvements in tools based on the use of cartoons for assessment after an emotion recognition program not found in photographs of real faces [34]. Moreover, using cartoons allows us to control the expressions of the emotions. It is widely recognized that cartoons have a strong advantage in expressing emotions and feelings. “Compared to human facial images, cartoon images have their own characteristics, i.e., firstly, the cartoon facial expressions can be extreme. Cartoons can show extreme expressions and extreme behavior without causing any negative feelings (e.g., the cartoon Simpson family’s behavior in TV shows). Secondly, parts of the characteristics presented in human images may be distorted or even disappear in cartoon images. For example, a cartoon image may have extremely large eyes without eyebrows or an upper lip.” [35]. However, there are no experiences in the assessment of emotion recognition using cartoons in people with schizophrenia. We hypothesized that an interactive gaming tool based on cartoon faces could be used in facial emotion recognition by people with schizophrenia for psychotherapeutic purposes.

Thus, the goal of the current study was to assess the usability, adaptability, and validation of ‘Feeling Master’, a psychotherapeutic interactive gaming tool for the assessment of emotional recognition in people with schizophrenia compared with healthy people. Moreover, we aimed to relate the results of emotional recognition to the attributional style and ToM.

## 2. Materials and Methods

### 2.1. Designing Facial Expressions Using Cartoon Techniques and ‘Feeling Master’ Program

The cartoon characters of ‘Feeling Master’ were created by an engineer from the Universitat Politècnica de Catalunya (UPC). To make simpler the development progress of our cartoon characters, the famous LEGO Minifigures illustrations were used. The LEGO minifigures consist of a head, torso, arms, hands, hip, and legs. A minifigure can have accessories in different parts of the minifigure like its head (such as hair, helmets, and hats) and around their neck (such as capes). There are a great number of possibilities to combine the parts, which allows us to provide an enormous variety of characters for our game.

The strongest indicator of the emotional state of the character is based on the face of the minifigure. The characteristics of the face and prominent facial features such as the eyes, eyebrows, and mouth are precise, as the detection of the position is the most important towards automatic recognition of facial expressions [36]. LEGO gave us the possibility to customize the character’s facial expression in a simple manner, considering facial features such as the eyes, eyebrows, and mouth.

In order to create expressive cartoons for the six basic emotions [37], we considered the following ground rules for customizing the character’s facial expressions: (1) happiness: mouth wide with the corners pulled up toward the ears and the eyebrows relaxed; (2) sadness: mouth relaxed and the inner portions of the eyebrows piled up above the upper eyelid; (3) fear: mouth dropped slightly open, eyebrows raised, pulled together, and bent upward; (4) surprise: mouth dropped open, eyebrows raised up, and upper eyelids opened; (5) disgust: mouth slightly opened with the upper lip squared off and the middle eyebrows pulled upward; and (6) anger: mouth closed with the upper lip slightly compressed or squared off, eyebrows pulled downward and together. These rules were summarized and simplified adapted from Parke and Waters’ book [38].

The customization process of the characters’ facial expressions starts with a small number of hand-drawn pencil sketches for the six basic facial expressions. An illustrator changed the facial expression of the original character illustrations considering three characteristics: eyes, eyebrows, and mouth, and based on the rules outlined above (Figure 1a). The changes to the original vector illustrations were applied once the pencil sketches were done for each facial expression (Figure 1b). An upper torso and arms were added to the illustration (Figure 1c). All the minifigure illustrations used during this process were developed by LEGO artists to be used in the company’s digital portfolio (websites, video games, and applications).

During the customization process of the character’s facial expressions, the same character was used to illustrate the six different emotions. This initial character was tested by a small group of users (seven participants) (Figure 1d). The users’ feedback was noted and analyzed, and then the illustrations were refined. The user testing cycle was repeated three times until the six different facial expressions on the same character had been successfully recognized. We then started expanding the number of cartoon characters for the game with diverse features and accessories (beards, moustaches, scarves, bows, shirts, uniforms, dresses, capes, and hair) (Figure 1). Finally, two different skin colors were combined (yellow and dark brown) to develop the final eighty characters needed for the game.

‘Feeling Master’ is an interactive game with three levels of difficulty. There is a maximum time allowance for responding to all questions. Time may differ in each level, the practitioner being the one who customizes the time of the game for each participant. The design is modular, once the level has been chosen, the patient has to correctly match an emotion to the corresponding face. In addition, therapists can manage information regarding patients’ data such us patients’ performance, in order to configure future sessions. Taking into account the abilities of the patient, the therapist could start with an easier or more complex activity, and with more or less time dedicated to identifying emotions. However, in this study, all patients were evaluated in the same way.

All, the correct and wrong responses were automatically recorded by the program, correlating with the emotion the user had been asked about in each question. Moreover, the total time dedicated by the user in the session and the score he or she attained, is also recorded. This information feeds into summary tables, with the percentages of right and wrong answers that are instantly available for the practitioner. Thus, the practitioner has an immediate feedback about the exactly correct or failed answer to discriminate a particular emotion and time spent to complete each level.

While designing the study, a prototype of the ‘Feeling Master’ application was revised in different meetings by different mental health professionals (psychologists from Parc Sanitari Sant Joan de Déu (PSSJD) and engineers from the Polytechnic University of Catalonia (UPC)). The application variables for this study, as number of questions, maximum time per question, and the selection of emotions, were defined during by the research team in these meetings and tested in pre-study trials. In a pre-pilot study, a total of five people with schizophrenia tested the tool, describing the program as simple to use, undemanding, and well accepted. Patients needed approximately 15 min to complete all three levels of the program. The patient participation was excellent and enjoyed all the sessions. This information was based on an interview with each participant made by the research team.

For the final study, the Flash application ‘Feeling Master’ was installed on a Samsung Wi-Fi Galaxy Tab 10.1-inch tablet powered by Android 3.1 Honeycomb operating system.

### 2.2. Content Validity and Reliability

An evaluation made by the team of designers was made in the design process to determine whether the cartoons were convenient. In the design process, the majority of users of the engineering staff (not directly involved in the development of the ‘Feeling Master’) had successfully recognized the six different facial expressions on the same character. Only disgust showed a high number of errors, and for this reason this emotion was eliminated from the last ‘Feeling Master’ version, yielding a final study with five emotions: happiness, sadness, anger, fear, and surprise.

Reliability and construct validity were evaluated comparing a total of seven designs, and 80 alternative designs were assessed by the engineering staff. The reliability was assessed including all the items regarding each emotion with Cronbach’s Alpha. The scores were up to 0.8 for each emotion assessed.

The levels of the game permitted a progressive introduction of the emotional recognition difficulty, in this way sustaining the motivation of the participants during the whole session.

### 2.3. Adaptability

Some tool adaptations were introduced in the interaction device, the learning phase, and the introduction of three game levels. ‘Feeling Master’ design took into account interactive devices, allowing for a touch screen. A learning phase based on a tutorial explanation was introduced.

The three levels of difficulty have the aim to adapt the game to the cognitive level of the participants. As commented previously, the variables for this study were tested in pre-study trials.

### 2.4. Participants

The study was conducted among participants with schizophrenia in five psychiatric rehabilitation services of PSSJD in Barcelona.

A total of 19 participants with schizophrenia participated in the study. The inclusion criteria for the participants with schizophrenia in the study were: DSM IV-TR criteria (APA, 2000) for schizophrenia, age between 18–45 years old, completion of high school education, and IQ score ≥ 85 (estimated using WAIS-IV [39]). A total of 13 patients were male (68.4% of the sample). The sample was composed of 17 single individuals (89.5% of the sample), 1 who was divorced (5.3% of the sample), and 1 widower (5.3%). Mean age of patients was 36.53 (SD = 7.10) with a mean of years of illness evolution of 13.14 (SD = 7.67). Number of hospitalizations throughout life was 2.53 (SD = 1.87). All patients were taking antipsychotic medication, 13 of them atypical antipsychotics.

The healthy control group was composed of 17 participants that were recruited to match the patient cohort at a group level in terms of age, gender, and education level. A total of nine participants were male (52.9% of the sample). Regarding marital status, nine participants were single (52.9% of the sample), five were married (29.4%), and three were divorced (17.6%). Average age of the control group was 35.29 (SD = 11.58). No gender or age differences were found between the participants with schizophrenia and the healthy control group.

### 2.5. Ethical Aspects

The Ethics Committee of PSSJD revised and approved the present study’s protocol. Participants were selected and contacted through the referring clinical psychologist of the PSSJD team. The study information was given to the participants, and they signed an informed consent to participate in the study.

### 2.6. Assessment


-Personal and Situational Attribution Questionnaire (IPSAQ) assesses the attributional style [40]. The IPSAQ consists of 32 items and each item describes positive and negative social situations. For each item, the participants are requested to identify the most likely cause of the event and then indicate whether that cause was attributed to oneself (internal attributions), other people (personal attributions), or circumstances (situational attributions). From the scores, it is possible to identify two cognitive biases: the externalizing bias (EB) and the personalizing bias (PB).-Theory of Mind: (ToM) is assessed using the Hinting Task [41], which consists of five written short stories, including social hints that the participants have to guess and explain. Total scores range from 0 to 10, higher scores indicate a better performance.-‘Feeling Master’, an interactive gaming tool used in facial emotion recognition: The interactive software application was used to assess facial emotion recognition (FER). When the application was installed and tested on the tablet, a trained psychologist adapted the number of questions for all of the three levels (seven questions), the highest amount of time per question (12 s), and the emotions needed for the FER evaluation (happiness, sadness, anger, fear, and surprise) (Figure 2).


The total number of questions for each game is 21. The levels are as follows. Level 1: select an emotion text description according with a cartoon (ex. How does this person feel? (a) happy, (b)) sad, or ((c) disgusted). Level 2: Select one emotion from three cartoon faces without accessories (ex. Who is happy? There are three possible answers according to the different facial expression of the same cartoon). Level 3: Select one emotion from three faces with accessories (ex. Who is happy? There are three possible answers according to the different facial expressions of the different cartoon customization), (Figure 3). The application indicates whether the answer is right or wrong with auditory and visual feedback.

The evaluation was made in two sessions, conducted on different days within a 1-week period. The evaluation was made individually with each patient in a quiet room in their psychiatric rehabilitation unit. Each session took about 10–15 min to finish and the administration was conducted by a psychologist and the application designer. The system recorded automatically all the participants response (right and wrong) for each game. The system also recorded the patient’s profile, the system configuration, the response time for each question, and the score obtained in every single game.

### 2.7. Statistical Analysis

Data were analyzed using PASW Statistics, version 18 (SPSS). The significance level was established at *p* = 0.05. The ANOVA was applied over accuracy (number of correct answers) and time needed to complete ‘Feeling Master’. Repeated ANOVA computation was used to analyze subject’s performance in the ‘Feeling Master’, per group (participants with schizophrenia and healthy control group). Fisher’s exact test was also used to compare error patterns between the participants with schizophrenia and the healthy control group. Spearman’s correlations were carried out in order to find out convergent validity between the emotion values achieved in FER and the IPSAQ and ToM scores for the participants with schizophrenia.

## 3. Results

### 3.1. Convergent Validity of the ‘Feeling Master’

A factor analysis and a correlation matrix approach were applied to examine the convergent and discriminant validity of the program considering the relationship of each image with the emotion represented. Summarizing, the inter-item correlations were happiness = 0.9, sadness = 0.9, fear = 0.8, surprise = 0.8, and anger = 0.8.

### 3.2. Acceptability and Usability

In relation to adaptability, results determine that 41 (100%) individuals in the sample were able to use a touch screen. Only two participants lost interest in the game, abandoning it between sessions on the same day (1) or on the second day (1).

### 3.3. Accuracy

Each of the five emotions was analyzed in terms of the ratings by all participants using the ‘Feeling Master’ tool. In the first session, the accuracy (correct answers in emotion recognition) range for each emotion was from 81.2–94.1% in the healthy control group, and from 67–88.5% among the participants with schizophrenia.

No significant values for overall emotion discrimination were found (average accuracy: F (1.38) = 0.733, *p* > 0.05), despite the fact a trend toward significance for error rates for discrimination of fear (F = (1.38) = 8.2, *p* = 0.07) was observed (Figure 4).

### 3.4. Response Time

The response time was significantly slower in the participants with schizophrenia in comparison with the control group. The mean time in the first session was 1.55 (SD = 0.57) minutes for the participants with schizophrenia and 0.92 (SD = 0.60) minutes for the healthy control group. The participants with schizophrenia continued being slower than the healthy control group over the sessions (Average time: F (1.37) = 15.1, *p* < 0.001).

### 3.5. Relationship between the Emotion Recognition Performance and IPSAQ and ToM

The correlations between correct response on the facial emotion recognition and IPSAQ and ToM are shown in Table 1. Only surprise and ToM were significantly associated (*p* = 0.046).

## 4. Discussion

### 4.1. Tool Design

‘Feeling Master’ is the first experiment in the assessment of emotional recognition in schizophrenia to consider an interactive software tool. The development of ‘Feeling Master’ was based on theoretical concepts and the experience of professionals. The pre-pilot study design was tested and found to be an easy tool to use. The results suggest that this technique could be useful for assessment of this area in this population. However, the level of difficulty used in the pilot study was low, and we were unable to find great differences between patients with schizophrenia and controls. The tool could be programmed at different levels of difficulty with limited response times; in future studies, the use of this tool could take into account these factors of difficulty in order to produce better discrimination.

### 4.2. Impairment in Emotional Recognition

Regarding the response time, patients spent more time identifying emotions than the control group. This result is congruent with the literature, in which several studies have found that patients with schizophrenia have problems with emotional recognition [7,8,10,11,12,13,14,42]. In our study, most of the patients recognized facial emotions. However, the latency in their response was greater. In contrast with other instruments of assessment of emotional recognition, ‘Feeling Master’ allows us to assess response time due to its interactive characteristics. We might consider that these differences found between controls and people with schizophrenia could be mediated by the effect of medication and cognitive deficits [43,44]. In any case, this latency in response is higher than in controls, and it might possibly affect their relationships in daily activities, although we could not determine the reaction time for each emotion, separately. Regarding emotion recognition, patients correctly recognized all the emotions presented in the facial emotion recognition except for fear, for which a trend toward significance was found. This result is consistent with other studies [10,14,17,45,46] in which fear was found to be the most impaired emotion in emotional recognition in patients with schizophrenia. Our results are in line with those of Adolphs [47], suggesting that negative faces are especially difficult to discriminate for people with schizophrenia, and that negative expressions are more difficult to recognize due to their shared characteristics with other expressions. Although in our study we only found differences in the emotion of fear, in the study of Romero-Ferreiro [45], the failure to recognize fear was present in people with a first episode of psychosis, indicating that this impairment could be an early marker of the disorder [48,49].

Surprisingly, in our study, only fear showed a tendency toward significant differences between patients and controls, while in most other studies, several emotions have been found to be affected. A point to take into account is that most of the studies that assess emotional recognition count on a limited time for the recognition of the emotion [45]. Therefore, patients with schizophrenia may not misattribute emotions but rather may have problems in the time allotted for emotional recognition.

### 4.3. Relation between Emotional Recognition, Attribution Style, and ToM

Fewer previous studies have related ToM, attributional style, and emotional recognition. Therefore, our results represent a novelty in this aspect. The study of Buck et al. [3] found that emotion recognition and ToM were in the same factor while attributional style was a different factor of social cognition. Our results are in this line, finding a relationship between ToM and emotional recognition of surprise but not with attributional style [50]. Regarding ToM, in our study, patients with lower scores in ToM presented difficulties in the recognition of surprise, in contrast with the study of Maat et al. [51], where they found that ToM are related with the failure to recognize angry faces. Studies have shown a relationship between impairment in ToM and emotional recognition, and more studies are needed in order to better outline the emotions involved, considering the inconsistencies between our study and Maat’s.

### 4.4. Future Work

This study has some limitations. Impaired emotion recognition in this study was limited to a group of patients with schizophrenia who were chronic and clinically stable. Other studies suggest that subject selection according to illness state or progression could affect emotional recognition [9,52,53,54]. Specifically, neutral faces should be included to test whether patients with schizophrenia misidentify neutral cues as unpleasant or threatening, relative to normal controls. The pattern of erroneous misattributions should be introduced for the number of recognized faces. In future studies, we should take into account the engagement of the patients in the assessment and interpretation of the images in order to improve the recognition of the emotion they express.

Another limitation of the study is the lack of information regarding psychotic symptoms. Future studies should include illness severity and its relation to emotional recognition, as well as the effect of medication, as suggested by several authors [14,55,56]. In future studies, we should also consider the influence of plasma oxytocin in emotion recognition.

In future studies, we will deliberately use time as a variable to reduce the accuracy in the responses in order to minimize the ceiling effect [57,58], which is high when a high value in scores is achieved for a large number of participants. Healthy subjects usually score at or near the maximum possible performance level in the ‘Feeling Master’ recognition test for all emotions. This situation is enhanced by using long stimulus exposure times that range from 500 milliseconds [52] to 15 s [59]. The ceiling effect was clearly observed when more than one session was introduced. One possible way to avoid the ceiling effect is by modifying the duration of stimulus presentation [57,58]. The improvement in the detection caused by the ceiling effect could help us to use this program as a learning tool for the improvement of emotional recognition. This program could be used in future psychosocial rehabilitation programs in order to measure progress in this ability and to train for it as well. There have been good studies using computer-based treatment or training in psychiatric patients with depression [60] and autism [34]. However, there is a lack of programs using computerized emotional recognition tasks in people with schizophrenia. Further investigation into computer-based treatment packages for emotion recognition remediation in patients with schizophrenia should be carried out.

In order to obtain statistically significant studies of ‘Feeling Master’, a larger sample of users must be assembled to determine the reliability and the validity of each cartoon in relation to the construct or the emotion that it represents.

In conclusion, the ‘Feeling Master’ tool is useful for assessing emotional recognition in people with schizophrenia. This application discriminates between healthy controls and people with schizophrenia regarding their impairment in emotion recognition, specifically for fearful faces.

## Figures and Tables

**Figure 1 jpm-11-00214-f001:**
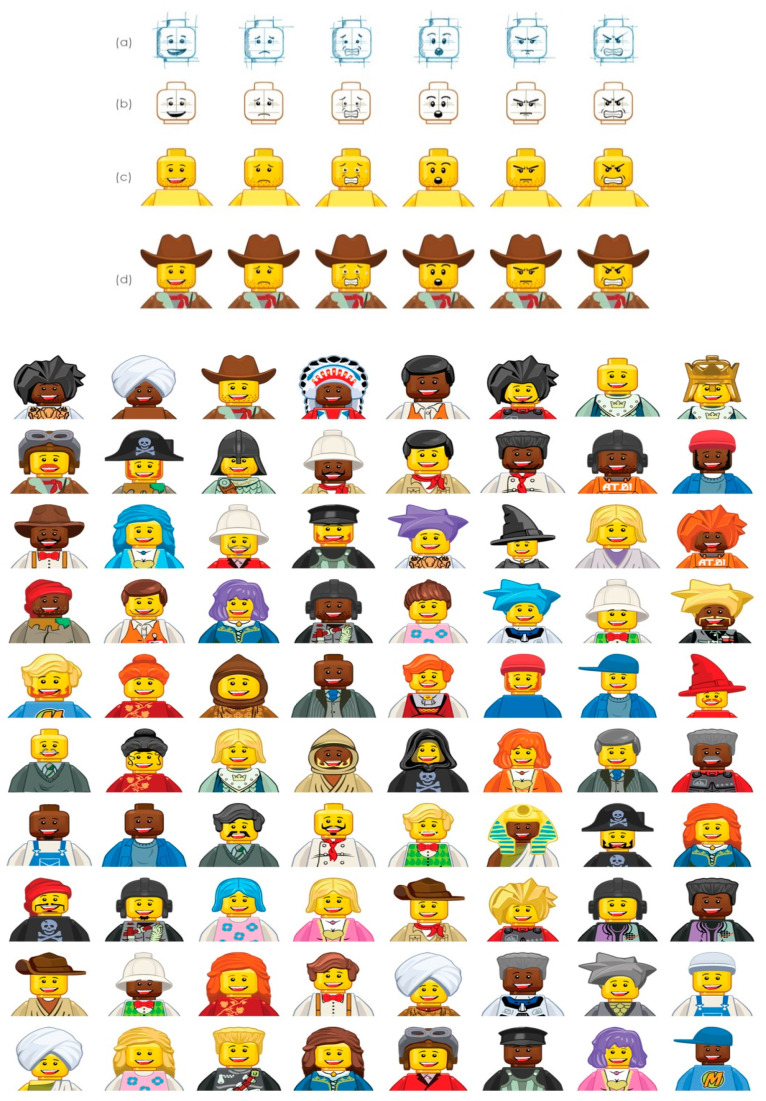
Facial expression design process: happiness, sadness, fear, surprise, disgust, and anger. (**a**) Pencil sketches. (**b**) Vector illustrations. (**c**) Minifigures for user testing. (**d**) Final characters.

**Figure 2 jpm-11-00214-f002:**
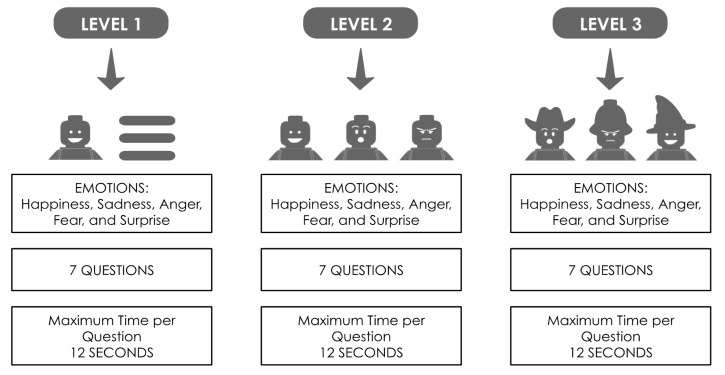
‘Feeling Master’ summary of levels.

**Figure 3 jpm-11-00214-f003:**
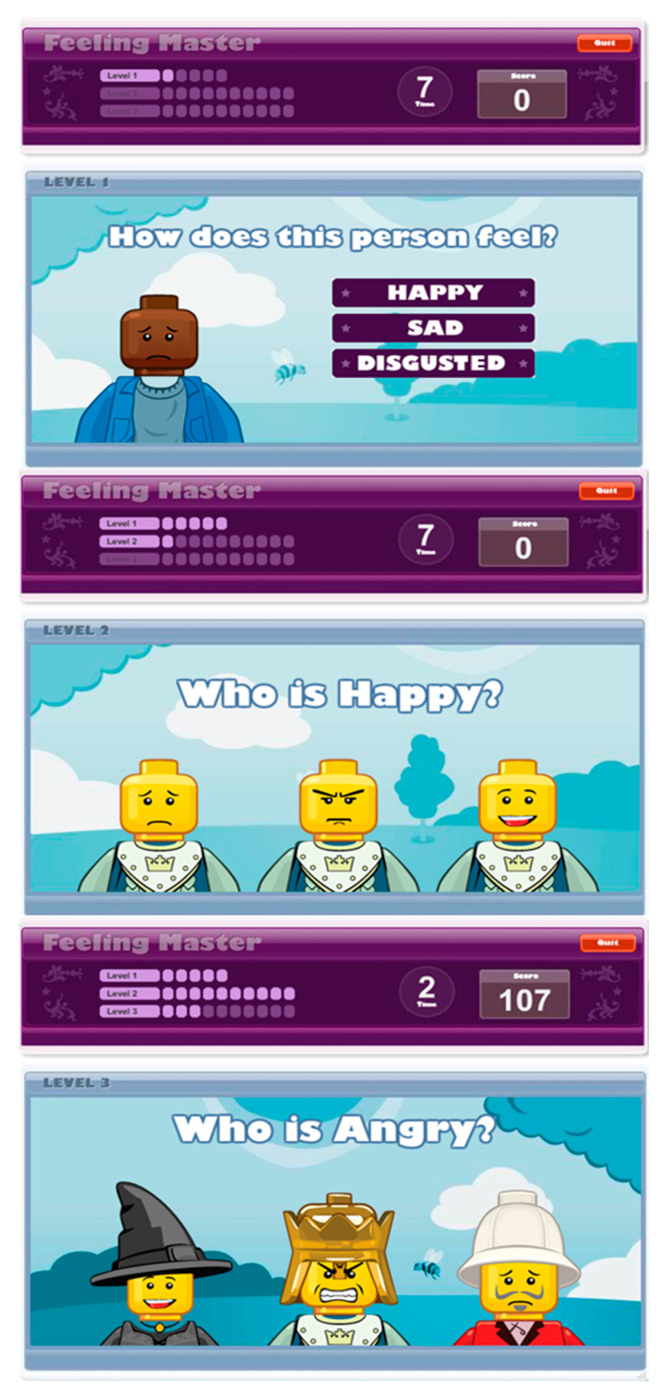
Interactive ‘Feeling Master’ tool.

**Figure 4 jpm-11-00214-f004:**
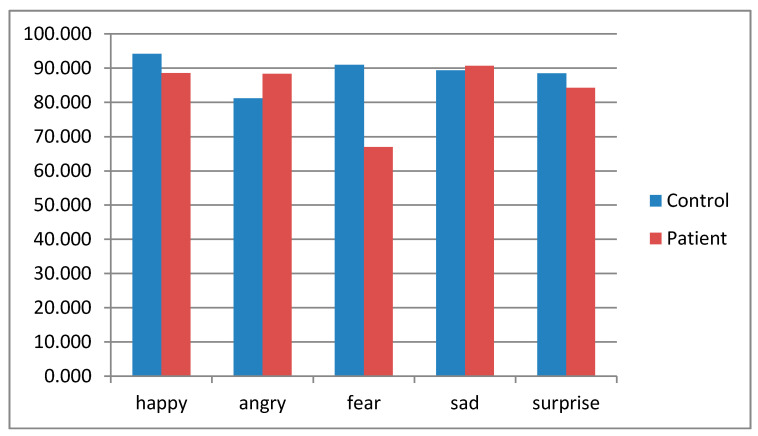
Percentage of correct answers in emotion recognition among patients with schizophrenia and the control group.

**Table 1 jpm-11-00214-t001:** Spearman correlations among IPSAQ, ToM, and facial recognition.

	IPSAQ_PB	IPSAQ EB	ToM
Happy Correlation coefficient Sig. (2-tailed)	−0.076	0.061	0.106
	0.730	0.776	0.630
Angry Correlation coefficient Sig. (2-tailed)	0.354	0.196	0.139
	0.115	0.383	0.547
Fear Correlation coefficient Sig. (2-tailed)	0.085	0.088	0.350
	0.700	0.676	0.102
Sad Correlation coefficient Sig. (2-tailed)	0.252	−0.162	0.261
	0.247	0.449	0.229
Surprise Correlation coefficient Sig. (2-tailed)	−0.028	0.182	0.430
	0.900	0.406	0.046

## Data Availability

Data is contained within this article.

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
