# Peer review of "Usability of a Psychotherapeutic Interactive Gaming Tool Used in Facial Emotion Recognition for People with Schizophrenia"

_jpm, 2021, doi:10.3390/jpm11030214_

Round 1

Reviewer 1 Report

This manuscript describes the development and evaluation for a new game, called "Feeling Master", in both healthy controls and patients with schizophrenia.

It is clear that the authors went through a long and thorough process in developing the software, and they clearly articulate the reasons for different aspects of the design process.

Overall, this is a high quality paper, and despite lack notable statistical findings. This software has many potential uses, and may prove more descriminating with new settings for stimulus presentations, as the authors suggest.

I have a few minor comments below:

Abstract

“FER” acronym not defined

What “Feeling master” does is unclear. What does it assess and how does it work?

IPSAQ is also not fully defined

If “HC” is going to be used for “Healthy Control”, it may also be appropriate to the use “SZ” for “Schizophrenia”, or don’t use these acronyms at all in the abstract.

ToM is not defined

In summary: The abstract needs a clearer definition of what the task actually involves, and correct many of the acronyms used.

Introduction

Line 41: This sentence is not correct: “Emotional recognition deficit is one of the components of social cognition.” It should be “Emotional recognition is one component of social cognition.”

Methods

Lines 107 – 109 should be removed

Author Response

Response to reviewer:

Reviewer 1

This manuscript describes the development and evaluation for a new game, called "Feeling Master", in both healthy controls and patients with schizophrenia.

It is clear that the authors went through a long and thorough process in developing the software, and they clearly articulate the reasons for different aspects of the design process.

Overall, this is a high quality paper, and despite lack notable statistical findings. This software has many potential uses, and may prove more descriminating with new settings for stimulus presentations, as the authors suggest.

We would like to acknowledge the reviewer for his/her comments. As the reviewer comment the development of the software was the main aim to describe in the study. The lack of findings in this preliminary data could be related with the fact of this study was a first approach to the program. We hope to perform further research in the future in order to better describe the tool and adapt it to our patients.

I have a few minor comments below:

Abstract

“FER” acronym not defined

What “Feeling master” does is unclear. What does it assess and how does it work?

IPSAQ is also not fully defined

If “HC” is going to be used for “Healthy Control”, it may also be appropriate to the use “SZ” for “Schizophrenia”, or don’t use these acronyms at all in the abstract.

ToM is not defined

In summary: The abstract needs a clearer definition of what the task actually involves, and correct many of the acronyms used.

Thank you very much for your comments. 

As the reviewer comment we have modified the acronyms that were not correctly identified in the abstract. We have modified the abstract and added information more clarifying about the FER acronym, feeling master tool, and better defined IPSAQ, ToM and HC and SZ.

Introduction

Line 41: This sentence is not correct: “Emotional recognition deficit is one of the components of social cognition.” It should be “Emotional recognition is one component of social cognition.”

Thank you very much for this comment. We have modified considering the reviewer’s suggestion

Methods

Lines 107 – 109 should be removed

Thank you very much for the detection of this error. We have removed from the manuscript.

Reviewer 2 Report

The aims of this study are not entirely clear. It seems to present a clear approach to assessing social cognition in schizophrenic patients. But it implies that the gaming approach taken will in itself offer a training benefit. but this is not clearly stated and if this is the case why was there no attempt to monitor  change in scores after exposure to the "training".no details of the patients is given eg age and ethnicity.Were they receiving pharmacological treatment? if so what?It is known that clozapine is effective in enhancing TOM  and the efficacy of clozapine is associated with carrying single nucleotide polymorphisms in the oxytocin receptor. the work would have been much more valuable had the authors examined plasma oxytocin levels and the influence of oxytocin receptor snps on test performance. There is some evidence that use of social media can influence plasma oxytocin

Author Response

Reviewer 2

The aims of this study are not entirely clear. It seems to present a clear approach to assessing social cognition in schizophrenic patients. But it implies that the gaming approach taken will in itself offer a training benefit. but this is not clearly stated and if this is the case why was there no attempt to monitor change in scores after exposure to the "training".no details of the patients is given eg age and ethnicity.

Thank you very much for your comments.

We agree totally with the reviewer’s comment about the aims of the study. It is true that they are not clear. We have re-written the aim of the study in order to clarify it and to describe the feeling master as a tool for the assessment of the emotional recognition. In future investigation we could assess the effectiveness this tool as a psychological intervention for emotional recognition, but it was not assessed in this way in the present study.

The modifications are:

“Thus, the goal of the current study was to assess the usability: adaptability and validation of ‘Feeling Master’, a psychotherapeutic interactive gaming tool for the assessment of emotional recognition in people with schizophrenia compared with healthy people. Moreover, we aimed to relate the results of emotional recognition to the attributional style and ToM.”

Regarding ethnicity, we did not mention it because it was not an important variable in our sample, done all the participants were Caucasians. We included the mean age of the participants in the method section.

Age information:

“Mean age of patients was 36.53 (SD=7.10) with a mean of years of illness evolution of 13.14 (SD=7.67. Number of hospitalizations throughout life was 2.53 (SD=1.87).”

Were they receiving pharmacological treatment? if so what?It is known that clozapine is effective in enhancing TOM  and the efficacy of clozapine is associated with carrying single nucleotide polymorphisms in the oxytocin receptor. the work would have been much more valuable had the authors examined plasma oxytocin levels and the influence of oxytocin receptor snps on test performance. There is some evidence that use of social media can influence plasma oxytocin

All the patients with schizophrenia were attended in rehabilitation services an all of them were taken medication. However, we have not collected detail information about specific pharmacological treatment, only the use of typical or atypical antipsychotics as we have mentioned in the manuscript. We have added this information as a limitation section of the study.

“Future studies should include illness severity and its relation to emotional recognition, as well as the effect of medication, as suggested by several authors [14,55,56].

We agree that the study of oxytocin is very interesting, but we had not the possibility to assess the oxytocin level in our patients.

Regarding the evidence of the influence of plasma oxytocin we have included a statement in the limitation section regarding this issue. For future studies we should consider the influence of oxytocin in emotional recognition.

“In future studies we should also consider the influence of plasma oxytocin in emotion recognition.” 

Round 2

Reviewer 2 Report

The paper is much improved, although I think the term Feeling Master is inappropriate and misleading. One thing that might be interesting would be to engage patients in manipulating the images.How do they make something that looks happy or sad?

Author Response

Reviewer response:

The paper is much improved, although I think the term Feeling Master is inappropriate and misleading.

Thank you very much for your comment.

We agree with you that the name feeling master could be misleading, but we made an agreement with LEGO with the condition of using this name and they also gave us permission to use the images with the name feeling master as well. We cannot change the name at this point because all the research has been made under the name feeling master.

One thing that might be interesting would be to engage patients in manipulating the images. How do they make something that looks happy or sad?

Thank you for your comment.

In future studies we could take into account the engagement of the patients in the assessment and interpretation of the images in order to improve the recognition of the emotion they express.

We included this recommendation in the future work section in lines 356 to 358.

Regarding your last question, within the lines 124 to 152 of the manuscript we describe the creation of each image and how it is made to express the different emotions.